# A Combined Analysis of Transcriptome and Proteome Reveals the Inhibitory Mechanism of a Novel Oligosaccharide Ester against *Penicillium italicum*

**DOI:** 10.3390/jof8020111

**Published:** 2022-01-25

**Authors:** Linyan Feng, Liangxiong Xu, Xiaojie Li, Jinghua Xue, Taotao Li, Xuewu Duan

**Affiliations:** 1Guangdong Provincial Key Laboratory of Applied Botany, South China Botanical Garden, Chinese Academy of Sciences, Guangzhou 510650, China; fenglinyannihao@163.com (L.F.); xuejh@scbg.ac.cn (J.X.); 2School of Life Sciences, Huizhou University, Huizhou 510607, China; xlx048@hzu.edu.cn (L.X.); LXJxiaojieli@163.com (X.L.); 3Agro-Food Science and Technology Research Institute, Guangxi Academy of Agricultural Sciences, Nanning 530007, China

**Keywords:** cell wall, gene, protein, citrus fruit, blue mold

## Abstract

Blue mold caused by *Penicillium italicum* is one of the most serious postharvest diseases of citrus fruit. The aim of this study was to investigate the inhibitory effect of a novel oligosaccharide ester, 6-*O*-*β*-L-mannopyranosyl-3-*O*-(2-methylbutanoyl)-4-*O*-(8-methyldecanoyl)-2-*O*-(4-methyl-hexanoyl) trehalose (MTE-1), against *P. italicum*. Scanning electron microscopy (SEM) and transmission electron microscopy (TEM), along with transcriptome and proteome analysis also, were conducted to illuminate the underlying mechanism. Results showed that MTE-1 significantly inhibited *P. italicum* growth in vitro in a dose-dependent manner. Moreover, MTE-1 suppressed the disease development of citrus fruit inoculated with *P. italicum*. Furthermore, ultrastructure observation, as well as transcriptome and proteome analysis, indicated that MTE-1 treatment damaged the cell wall and plasma membrane in spores and mycelia of *P. italicum*. In addition, MTE-1 regulated genes or proteins involved in primary metabolism, cell-wall metabolism, and pathogenicity. These results demonstrate that MTE-1 inhibited *P. italicum* by damaging cell walls and membranes and disrupting normal cellular metabolism. These findings contribute to the understanding of the possible molecular action of MTE-1. Finally, MTE-1 also provides a new natural strategy for controlling diseases in postharvest fruit.

## 1. Introduction

Fungal infection causes not only severe economic losses and fruit-quality deterioration during storage but also leads to adverse impacts on human health caused by mycotoxins [1,2]. *Penicillium italicum* is an important postharvest pathogenic fungus that causes blue mold on citrus fruit. It is difficult to control this pathogenic fungus because *P. italicum* is able to directly attack healthy fruit, regardless of injury [3]. Due to the severe spoilage of citrus fruit and severe economic loss caused by blue mold, additional efforts are urgently required to explore safe and efficient strategies to inhibit *P. italicum* growth and prevent infection of harvested citrus fruit. Although application of chemical fungicides is effective for controlling fungal disease, resistant fungal populations and environmental contamination, as well as human health, are increasingly attracting the concern of researchers [4]. In this regard, alternative strategies for fungicide application are imperatively required [5].

In recent years, natural bioactive compounds, such as different types of oligosaccharides, as opposed to synthetic chemicals, have received considerable attention for their potential to manage postharvest diseases [6,7]. Normally, functional oligosaccharides consist of two to ten monosaccharides that are linked together with glycosidic bonds [8]. Some oligosaccharides have been used on agricultural commodities, such as fruits and vegetables [9]. Additionally, oligosaccharides are commonly used as antimicrobial agents and have been considered as potential natural antimicrobial preservatives [10]. The preservation effects of different oligosaccharides on delaying ripening and senescence of fruits during postharvest storage, as well as its underlying mechanism, have been widely reported in recent years [7]. However, the effects of oligosaccharides or oligosaccharide esters on the growth of *P. italicum* and its possible mechanisms have not been studied.

A new oligosaccharide ester with antifungal potential, 6-*O*-*β*-L-mannopyranosyl-3-*O*-(2-methylbutanoyl)-4-*O*-(8-methyldecanoyl)-2-*O*-(4-methylhexanoyl) trehalose (MTE-1), was first found from cultures of *Pezicula neosporulosa* SC1337 [11]. Currently, there are no reports on the effect of MTE-1 on postharvest disease in fruits. In this study, we aimed to investigate the inhibitory effect of MTE-1 on *P. italicum* in vitro and in vivo. Furthermore, we conducted transcriptome and proteome analyses, accompanied by SEM and TEM observation of *P. italicum*, to explore the relevant antifungal mechanisms conferred by MTE-1 treatment.

## 2. Materials and Methods

### 2.1. MTE Preparation

6-*O*-*β-*L-mannopyranosyl-3-*O*-(2-methylbutanoyl)-4-*O*-(8-methyldecanoyl)-2-*O*-(4-methylhexanoyl) trehalose (MTE-1) was isolated from the cultures of *P. neosporulosa* SC1337 and identified by HRESI-MS and NMR [11].

### 2.2. Fungal Strain

*P. italicum* was originally isolated from decayed citrus fruit [12] and stored in 50% glycerol at −80 °C. Before use. *P. italicum* was cultured at 28 °C on potato dextrose agar (PDA) plates (Oxoid, Basingstoke, Hampshire, UK) for 7 days. Then, the spores were collected with sterile deionized water, filtered, and then counted with a hematocytometer for further experiments.

### 2.3. Antifungal Activity of MTE-1 against P. italicum In Vitro

Antifungal activity of MTE-1 against *P. italicum* was evaluated by paper-disc agar-diffusion method [13]. In brief, spore concentration was adjusted to 1 × 10^6^/mL, and spores we poured evenly on the PDA plate. Then, sterile filter-paper discs (4 mm) containing 0, 0.2, 0.4, and 0.6 g/L MTE-1 (30 μL), respectively, were gently pressed onto the surface of the agar plates. Sterile distilled water and thiabendazole were used as the negative and positive controls, respectively. The plates were incubated at 28 °C for 7 days, and the diameter of the inhibition zone was measured.

Antifungal efficacy of MTE-1 against *P. italicum* was evaluated by mycelial growth assay. PDA medium was used to grow fungal colonies. After 5 days of culture, agar discs (6 mm in diameter) with mycelia were excised and then transplanted to the center of PDA dishes (90 mm in diameter) with 0.5–20 mg/L MTE-1 under sterile conditions. The diameters of the colonies were measured to analyze the growth of *P. italicum*. The concentrations of EC50 were determined using GraphPad Prism software, version 7.00 (Graphpad Software, Inc., San Diego, CA, USA).

### 2.4. Effect of MTE-1 on Disease Development of Citrus Fruit Inoculated with P. italicum

Citrus fruit (*Citrus reticulata* Blanco) was harvested at about 70–80% maturation from a local orchard in Guangzhou City, China. Fruit was transferred to the laboratory within 2 h and selected with uniform size and color and with wounds. After sterilization with sodium hypochlorite (1%), the fruit was wounded with a sterile puncher. A total of 50-μL of MTE-1 at 0.4 g/L was injected into each wound. Sterile distilled water and thiabendazole were used as the negative and positive controls, respectively. Subsequently, 20 μL of conidium suspension (1 × 10^6^ CFU per milliliter) was injected into the wound. The fruit was then packaged in 0.02 mm polyethylene bags and incubated at 26 °C. After 7 days of incubation, the number of infected fruits was recorded, and the disease index was calculated.

The disease index was calculated using the following formula: disease index (%) = [(0 × n1 + 1 × n2 + 2 × n3 + 3 × n4 + 4 × n5 + 5 × n6 + 6 × n7 + 7 × n8 + 8 × n9 + 9 × n10)/(N × 9)] × 100. N, total number of the inoculation sites; n1, number of inoculation sites that developed no lesion; n2–n9, number of lesions with a diameter in the range of 0.5–1, 1–2, 2–3, 3–4, 4–5, 5–6, 6–7, or 7–8 cm, respectively; n10, number of lesions with a diameter >8 cm.

### 2.5. Scanning Electron Microscopy (SEM) and Transmission Electron Microscopy (TEM)

*P. italicum* was cultured on a potato-dextrose agar in 90-mm Petri dishes in the presence and absence of 3.69 mg/L (EC_50_) MTE-1 for 5 days at 28 °C. SEM and TEM analyses of *P. italicum* spores and hyphae morphology were conducted according to our previous approach [14].

### 2.6. RNA Extraction and Transcriptome Analysis

After incubation in PDB for 48 h, *P. italicum* was treated with 3.69 mg/L MTE-1 (EC_50_) for an additional 3 h. Then, the mycelia were collected for total RNA extraction using Hipure Fungal RNA Mini Kit (Magen, Shanghai, China). Then RNA was purified, checked, and sequenced according to our previous research [15]. Differentially expressed genes (DEGs) were identified with |log2Ratio| ≥ 1 (FDR < 0.05). Afterwards, GO enrichment with corrected *p*-value ≤ 0.05 and KEGG pathway enrichment with Q value ≤ 0.05 were also performed for DEGs.

### 2.7. Protein Extraction and Proteome Analysis

The same mycelium sample as described in 2.6 was used for protein extraction. Total proteins were extracted using around 500 mg mycelium samples from different treatments. In brief, samples were ground to power in liquid nitrogen, then dissolved in 2 mL lysis buffer (8 M urea, 2% SDS, 1× Protease Inhibitor Cocktail (Roche Ltd., Basel, Switzerland)), followed by sonication on ice for 30 min and centrifugation at 13,000 rpm for 30 min at 4 °C. The supernatant was transferred to a fresh tube. For each sample, proteins were precipitated with ice-cold acetone at −20 °C overnight. The precipitations were cleaned with acetone three times and redissolved in 8 M urea by sonication on ice. Protein quality was examined with SDS-PAGE, and the concentration was determined using BCA Protein Assay Kit. Then, proteins were tryptically digested with sequence-grade modified trypsin (Promega, Madison, WI). The digested samples were then dissolved in 500 mM TEAB and labeled with iTRAQ tags (iTRAQ Reagents-8Plex (SCIEX)). After high-pH reverse-phase separation, twelve fractions were collected for nano-HPLC-MS/MS analysis, which was performed on an Easy-nLC 1000 system (Thermo Fisher Scientific, Waltham, MA, USA) connected to an Orbitrap Fusion Tribrid mass spectrometer (Thermo Fisher Scientific, Waltham, MA, USA) equipped with an online nano-electrospray ion source. The fusion mass spectrometer was operated in data-dependent acquisition mode to switch automatically between MS and MS/MS acquisition.

Once transformed into MGF files by Proteome Discovery 1.2 (Thermo, Pittsburgh, PA, USA), the mass-spectrometry data were analyzed using Mascot search engine (Matrix Science, London, UK; version 2.3.2) and searched against Mascot database for protein identification using transcriptome data of *P. italicum*. Finally, protein quantification was carried out, and differentially accumulated proteins (DAPs) were determined with fold change in a comparison >1.2 or <0.83 (*p* < 0.05). All DAPs were further analyzed by GO and KEGG enrichment with a *p* value ≤ 0.05.

### 2.8. Data Analysis

All experiments were performed with three biological replicates. The data were represented as the means ± standard error (SE). Statistical analysis was conducted by SPSS (Version 20, SPSS Inc., Chicago, IL, USA).

## 3. Results

### 3.1. Effect of MTE-1 Treatment on the Growth of P. italicum

As shown in Figure 1A, MTE-1 obviously inhibited mycelium growth of *P. italicum*. The inhibitory effect increased with increasing concentration. At high concentrations, the inhibition efficiency of MTE-1 against *P. italicum* was slightly higher than that of thiabendazole, a commercial fungicide. At 0.4 g/L, the diameters of the MTE-1 and thiabendazole inhibition zones were 51.6 mm and 47.4 mm, respectively (Figure 1B).

Mycelial growth assay further showed that MTE-1 inhibited mycelium growth of *P. italicum* in a dose-dependent manner, with the EC_50_ of 3.69 mg/L. After 7 days of culture at 28 °C, the inhibition rate of hyphae growth was 88.3% for the 20 mg/L MTE-1 treatment (Figure 2).

### 3.2. Effect of MTE-1 on Disease Development in Mandarin Fruit Inoculated with P. italicum

To clarify the efficacy of MTE-1 in the inhibition of blue mold, we treated the harvested citrus fruit with MTE-1 before pathogen inoculation. As shown in Figure 3, the disease index of blue mold was 51.8% after 15 days of storage at 25 °C. MTE-1 treatments significantly inhibited blue-mold development (Figure 3A). When a concentration of 0.4 g/L was applied, the disease indexes were 23.1% and 24.7% in MTE-1- and TBZ-treated fruits, respectively (Figure 3A). Moreover, the inhibitory effect MTE-1 on blue-mold development was comparable to that of TBZ (Figure 3A,B).

### 3.3. SEM and TEM Analysis

SEM and TEM observations were conducted to explore the morphology and ultrastructural alteration of *P. italicum* caused by MTE-1 application. SEM observation revealed that the mycelium control group showed normal morphology (Figure 4A), while mycelia from the MTE-1-treated group exhibited deformed structure, with more conidiophores than the control (Figure 4B). Meanwhile, the number of spores was evidently reduced after MTE-1 treatment compared to the control group (Figure 4B). Additionally, TEM observation was conducted to further explain the inhibitory mechanism of the MTE-1 treatment. TEM results showed that the ultrastructure of the *P. italicum* had an intact plasmalemma system and cell wall in the absence of MTE-1 (Figure 4C). In contrast, the general cell ultrastructure of *P. italicum* was modified after MTE-1 treatment. Disruption of cell walls was evident in MTE-1-treated *P. italicum*, indicated by a coarse cell-wall surface and loss of rigidity and integrity of cell walls after treatment. (Figure 4D). Moreover, many transparent inclusions with wide vacuoles were observed in MTE-1-treated cells (Figure 4D), indicating leakage of intracellular substances. Altogether, SEM and TEM results reveal that MTE-1 damages mycelia morphology and spore ultrastructure of *P. italicum*.

### 3.4. Transcriptome Analysis of P. italicum in Response to MTE-1

To obtain the global changes in genes regulated by MTE-1 treatment, transcriptomes of the mycelium samples were analyzed. In response to MTE-1 treatment, 1226 genes were differentially expressed, with 984 upregulated and 242 downregulated (Figure 5A).

To investigate the biological functions of these DEGs, GO enrichment analysis was performed. Table 1 shows the TOP 20 enriched GO items based on biological process. Meanwhile, according to cellular-component (CC) category, DEGs belonging to “membrane” and “membrane part” were enriched (Figure 5B).

Furthermore, KEGG enrichment was conducted. Among the top 20 enriched pathways, most of the KEGG terms belong to amino-acid metabolism (Table 2). In addition, we identified significantly enriched metabolic pathways, including fructose and mannose metabolism, etc., using KEGG enrichment analysis (Table 2). Furthermore, genes involved in glycolysis, the tricarboxylic acid (TCA) cycle, and lipid metabolism were also active after MTE-1 treatment (Figure 6A–C, Appendix A).

### 3.5. Proteome Analysis of P. italicum in Response to MTE-1

iTRAQ-based proteome analysis was conducted to investigate the effect of MTE-1 on protein accumulation in *P. italicum*. As shown in Figure 5C, unlike DEGs, only 70 proteins were influenced by MTE-1, with 48 upregulated and 22 downregulated (Figure 5C).

In GO analysis, similarly to transcriptome data, DAPs belonging to “membrane” and “membrane part” were found according to cellular-component (CC) category (Figure 5D, Appendix A).

Similarly to transcriptomic data, ROS-related proteins (PITC_069590, PITC_045980 and PITC_090850) were upregulated in response to MTE-1 application. Downregulation of PITC_004590 (mannitol dehydrogenase) and PITC_025750 (ankyrin repeat-containing domain) was observed in the present study (Figure 6D).

## 4. Discussion

Blue mold caused by *P. italicum*, which causes postharvest fruit spoilage, is one of the most economically important diseases [16]. Our previous research identified a series of novel 6-*O*-*β*-L-manno-pyranosyl trehalose esters (MTEs), including MTE-1, exhibiting high inhibitory activity against many kinds of plant pathogenic fungi [11,17]. Hence, in this study, we investigated the effect of MTE-1 on the growth of *P. italicum* in vitro and in vivo. Transcriptome and proteome analyses were also conducted to study the underlying mechanism at the molecular level.

### 4.1. MTE-1 Treatment Inhibited P. italicum Growth

A previous study demonstrated that alginate oligosaccharide is effective in inhibition of blue mold on citrus fruit [18]. In the current study, MTE-1 was also proven to exhibit high inhibitory activity against *P. italicum* growth both in vitro and in vivo in a dose-dependent manner (Figure 1 and Figure 2). Additionally, in vivo assay indicated that MTE-1 exhibited inhibitory efficacy against *P. italicum* growth on citrus fruit. The results obtained in this study reveal that MTE-1 could be a potential treatment for inhibition of *P. italicum* in postharvest fields.

Primary metabolism is vital for fungal growth. Respiration is driven by a number of biochemical pathways, including the tricarboxylic acid (TCA) cycle, and consumes metabolic substances, leading to cell senescence. In this study, all genes involved in the tricarboxylic acid (TCA) cycle (PITC_028840, PITC_046710, PITC_075470, PITC_093040, and PITC_090210) were upregulated. In contrast, phosphoenolpyruvate carboxykinase (PITC_095870), which plays a vital role in intracellular carbon-skeleton recycling [19], was downregulated by MTE-1 treatment. We proposed that MTE-1 treatment might accelerate the consumption of substances but inhibit the recycling of carbon skeletons, thereby inhibiting the growth of *P. italicum*. In line with our results, a prior study showed that proteins involved in the pyruvate metabolic process and the TCA cycle are markedly induced in response to antifungal agents [20]. Interestingly, proteomic results showed that the accumulation of bifunctional 6-phosphofructo-2-kinase/fructose-2, 6-bisphosphate 2-phosphatase (PITC_080350) was significantly downregulated by MTE-1, suggesting the inhibition of the glycolytic pathway. Mitochondrial carriers provide a link between metabolic reactions occurring in the cytosol and the mitochondrial matrix and are involved in many important metabolic pathways [21]. In this study, mitochondrial substrate/solute carrier (PITC_016450) was downregulated by MTE-1, suggesting the dysfunction of mitochondrial amino-acid metabolism, especially branched-chain amino-acid metabolism, which also contributes to pyruvate metabolism and the TCA cycle [22]. In this study, genes involved in valine, leucine, and isoleucine degradation were significantly upregulated in MTE-1-treated *P. italicum*. Based on these results, propose that the alteration of primary metabolism pathways caused by MTE-1 might result in the inhibition of *P. italicum* growth.

### 4.2. MTE-1 Treatment Disrupted the Cell Structure of P. italicum

The damage of cell structure was evidenced by SEM and TEM observations in the present study. SEM observations showed that mycelia of the pathogen treated with MTE-1 appeared significantly deformed with conidiophores. According to TEM observations, the cell walls and plasmalemma systems of spores were altered by MTE-1, and plasmolysis was observed. These alterations are in agreement with the obsevered effects of other fungicides applied on different fungi [23]. Similarly, the application of chitosan to a variety fungi also results in mycelial aggregation and morphological-structural changes, such as abnormal shapes and hyphal swelling [24,25]. MTE-1 treatment resulted in the disturbance in the morphogenesis and growth of *P. italicum*, which is in agreement with the previous reports [26].

It is well reported that antifungal drugs can target an impose severe stress on the fungal cell wall [27]. For example, a recent study found that aseptic filtrate significantly improved crucial-cell wall enzyme activity to damage cell walls of the *Alternaria alternata* [28]. Fungal β-1, 3-glucanase are distributed in different glycoside hydrolase (GH) families and plays an important role in cell-wall remodeling and modifications [29]. Li et al. [28] found that β-1, 3-glucanase activity prominently increases in *A. alternata* after aseptic filtrate treatment, resulting in serious cell-wall injury. In this study, MTE-1 treatment significantly upregulated *glycoside hydrolase* (PITC_043150) expression, suggesting an accelerating effect of MTE-1 on cell-wall damage.

It was reported that membrane modification occurs mainly through interactions with phospholipids of the plasma membrane [30]. Phospholipase D is an important membrane-lipid-degrading enzyme that promotes the disruption of the cell membrane [31]. Similarly, our results show that MTE-1 induced the expression of *phospholipase D* (PITC_052500) compared to the control. The upregulation of this gene might contribute to the disruption of plasma-membrane integrity of *P. italicum*, which is consistent with TEM results. In agreement with SEM and TEM analyses, both transcriptomic and proteomic analyses showed that the plasma membrane was a target for the inhibitory action of MTE-1. Chen et al. [32] reported that damage to the lipid bilayer of the membrane results in cellular collapse. In this study, lipid metabolism was found to be rather active in response to MTE-1 treatment, which suggests that the disruption of the *P. italicum* membrane by MTE-1 might be related to lipid-biosynthesis disorders. In all, our results demonstrate that ultrastructural alteration is one of the antifungal mechanisms of MTE-1, which is consistent with previous research on reported bioactive compounds [14,33].

### 4.3. MTE-1 Treatment Regulated the Stress Response of P. italicum

In response to MTE-1 treatment, ROS stress-related genes, such as *catalase* (PITC_090850) and *glutathione reductase* (PITC_044110), were upregulated in *P. italicum*. Similarly, proteomic data also showed upregulation of ROS-related proteins (PITC_090850, PITC_045980, PITC_069590). These results suggest that oxidative stress was induced by MTE-1 treatment, which might activate an oxidative response in *P. italicum*. In fact, a similar mechanism was also observed in other antifungal compounds [34]. In addition, other genes falling into the “response to stress” were all upregulated after MTE-1 treatment. It is worth noting that two nitroreductase proteins (PITC_008180 and PITC_054850) were identified as upregulated in MTE-1-treated sample. Similarly, in *Metarhizium robertsii*, nitroreductase-like proteins were found to be engaged in the removal of the xenobiotic in response to a 4-n-nonylphenol (4-n-NP) pollutant [35]. All these results suggest that MTE-1 can cause obvious stress to *P. italicum*, affecting growth.

### 4.4. MTE-1 Treatment Affected the Pathogenicity of P. italicum on Citrus Fruit

In this study, MTE-1 treatment inhibited the virulence of *P. italicum* on citrus fruit, indicated by the inhibition of lesion development of blue mold after MTE-1 treatment. Importantly, transcriptomic and proteomic analyses also identified differentially expressed genes or proteins after MTE-1 treatment. Ankyrin repeat-containing protein (ANK) was reported to represent a new family of bacterial type IV effectors that play a major role in host-pathogen interactions and the evolution of infections [36]. In this study, ankyrin repeat-containing domain protein was identified in our proteomic results, with significant downregulation by MTE-1. The exact role of ankyrin repeat-containing domain protein (PITC_025750) in *P. italicum* needs further investigation. Adherence to host tissues plays a vital role in fungal infection [37]. Two hydrophobins (PITC_015600 and PITC_077120) were also identified, with downregulation by MTE-1 at both gene and protein levels. A prior study suggested the involvement of a *hydrophobin* gene with fungal pathogenesis through regulation of the formation of appressoria, which is required for infection of the host [37]. ROS produced by phytopathogenic fungi play crucial roles in the regulation of fungal virulence [38]. NCF2 is a component of the NADPH oxidase complex that produces extracellular ROS and mutations in NCF2, leading to decreased ROS production [39]. Particularly, the subunits of the NOX complex are well known to affect the virulence of phytopathogenic fungi via mediation of the differentiation and development of specialized infection structures of the fungi (reactive oxygen species: a generalist in regulation of development and pathogenicity of phytopathogenic fungi). Our proteomic results show that neutrophil cytosol factor 2 (PITC_083450) was downregulated after MTE-1 treatment, which might contribute to inhibition of disease development by MTE-1 on citrus fruit. In response to pathogen infection, plant hosts also produce ROS for defense. During pathogen-plant interaction, mannitol in fungi can detoxify ROS generated in the host environment and protect the plant’s ROS-mediated defenses, suggesting mannitol as a pathogenicity factor in fungi [40]. Mannitol dehydrogenase (MTD) catabolizes the biosynthesis of mannitol, and mannitol accumulation is paralleled with high levels of fungal MTD activity (fungal mannitol biosynthesis) in the haustoria [41]. In this study, mannitol dehydrogenase (PITC_004590) accumulation was reduced by MTE-1 treatment. Altogether, these results lead us to postulate that MTE-1 reduced the virulence of *P. italicum*. Certainly, the imbalance of intracellular and extracellular ROS was not clear in response to MTE-1 treatment.

## 5. Conclusions

In sum, we found that MTE-1 exerted high inhibitory activity against *P. italicum* in vitro and decreased the disease incidence of blue mold on postharvest citrus fruit during storage. Further, SEM and TEM observations, along with transcriptomic and proteomic results, revealed that treatment with MTE-1 disrupted cell walls and ultrastructural membranes, which might account for the growth inhibition. In addition, transcriptomic and proteomic analyses suggested that MTE-1 treatment might alter primary metabolism and confer stress on *P. italicum*, resulting growth inhibition. Importantly, MTE-1 affected the pathogenicity of *P. italicum* by regulating virulence-related genes or proteins. We conclude that MTE-1 treatment might be an effective strategy for controlling postharvest decay in citrus fruit.

## Figures and Tables

**Figure 1 jof-08-00111-f001:**
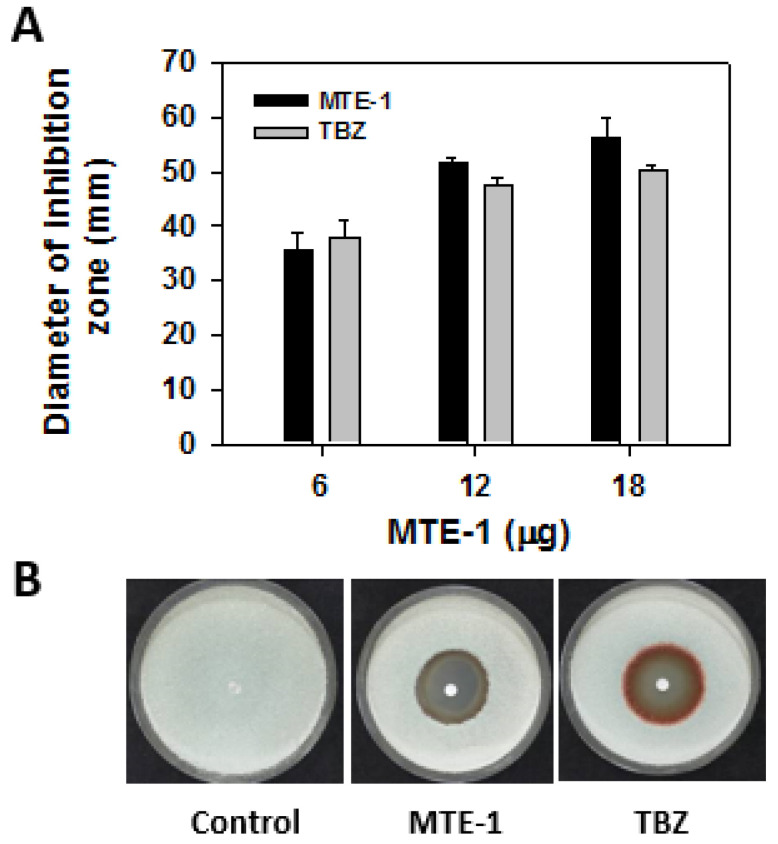
Effect of MTE-1 on mycelium growth of *Penicillium italicum* in vitro. (**A**) Diameters of inhibition zones of filter-paper discs (4 mm) containing 6 μg, 12 μg, and 18 μg MTE-1 (30 μL) after 3 days of incubation at 28 °C. (**B**) Colony morphology of *Penicillium italicum* on PDA plates with filter-paper discs containing 12 μg MTE-1 after 3 days of incubation at 28 °C.

**Figure 2 jof-08-00111-f002:**
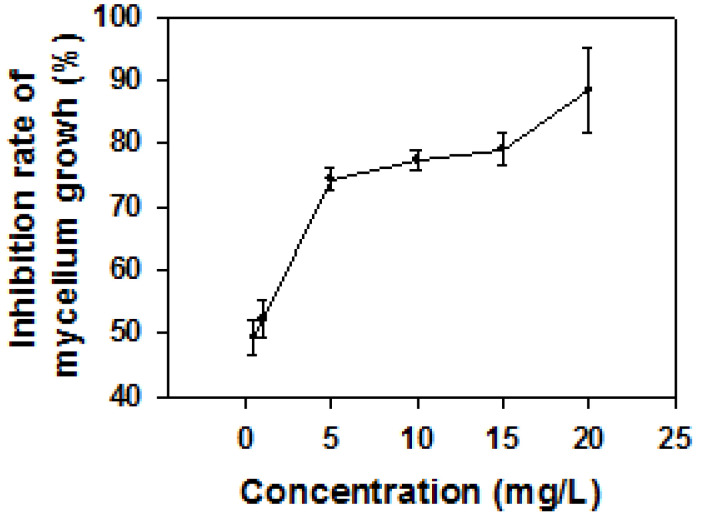
Antifungal activity of MTE-1 against *Penicillium italicum* growth in vitro.

**Figure 3 jof-08-00111-f003:**
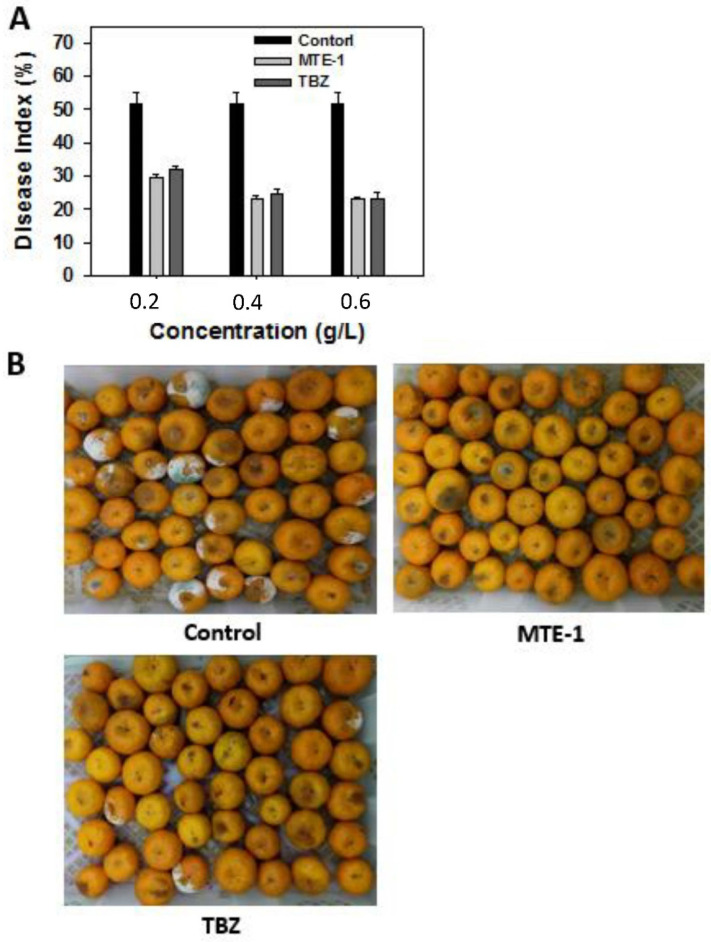
Effect of MTE-1 on disease development of mandarin fruit inoculated with *Penicillium italicum*. (**A**) Disease indexes of blue mold in mandarin fruit inoculated with *Penicillium italicum* after 15 days of storage at 25 °C. (**B**) Appearance of mandarin fruit treated with 0.4 g/L MTE-1 and subsequently inoculated with *Penicillium italicum* after 15 days of storage at 25 °C.

**Figure 4 jof-08-00111-f004:**
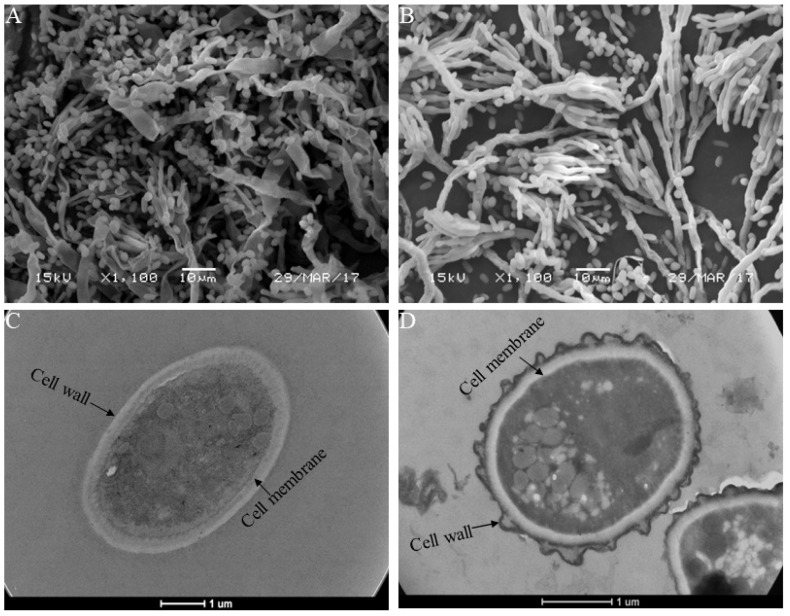
Ultrastructure changes in *Penicillium italicum* after MTE-1 treatment. (**A**) SEM image of mycelia in the control sample; (**B**) SEM image of mycelia in the MTE-1-treated sample; (**C**) TEM image of a spore in the control sample; (**D**) TEM image of a spore in the MTE-1-treated sample.

**Figure 5 jof-08-00111-f005:**
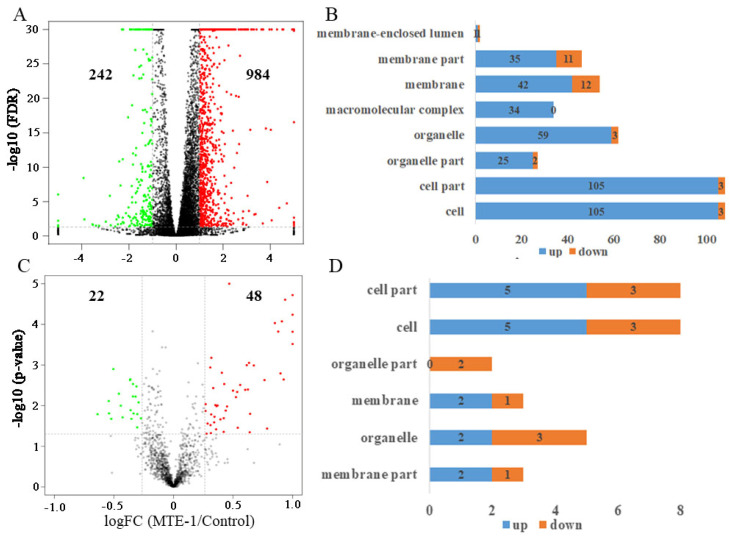
(**A**) Volcano plot of differentially expressed genes (DEGs) in *Penicillium italicum*; (**B**) function analysis of DEGs based on cellular-component category; (**C**) volcano plot of differentially accumulated proteins (DAPs) in *Penicillium italicum*; (**D**) function analysis of DAPs based on cellular-component category.

**Figure 6 jof-08-00111-f006:**
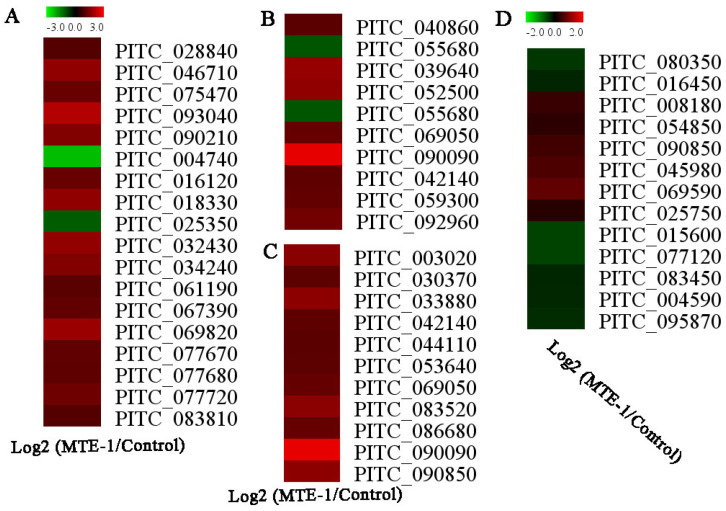
Selected genes or proteins in *Penicillium italicum* that were differentially regulated by MTE-1. (**A**) Genes involved in primary metabolism; (**B**) genes involved in cell-wall and lipid metabolism; (**C**) genes involved in response to stress; (**D**) differentially regulated proteins.

**Table 1 jof-08-00111-t001:** GO enrichment of DEGs in *Penicillium italicum*.

GO ID	Description	Gene Number	*p*-Value
GO:0018208	peptidyl-proline modification	6	0.000379783
GO:0006520	cellular amino-acid metabolic process	29	0.000873782
GO:0044710	single-organism metabolic process	145	0.001147218
GO:0019752	carboxylic-acid metabolic process	37	0.002046061
GO:0006082	organic acid metabolic process	37	0.003157122
GO:0043436	oxoacid metabolic process	37	0.003157122
GO:0006790	sulfur-compound metabolic process	10	0.003760795
GO:0009081	branched-chain amino-acid metabolic process	5	0.003761958
GO:0044699	single-organism process	201	0.012854284
GO:1901605	alpha-amino-acid metabolic process	15	0.013183352
GO:0018193	peptidyl-amino-acid modification	6	0.015477395
GO:0006549	isoleucine metabolic process	3	0.020871202
GO:0072525	pyridine-containing compound biosynthetic process	3	0.020871202
GO:0000096	sulfur amino-acid metabolic process	6	0.02121723
GO:0006733	oxidoreduction coenzyme metabolic process	4	0.025168398
GO:0044281	small-molecule metabolic process	66	0.030835746
GO:0009066	aspartate-family amino-acid metabolic process	5	0.033230704
GO:0044272	sulfur-compound biosynthetic process	5	0.033230704
GO:1901564	organonitrogen-compound metabolic process	56	0.037409962
GO:0006007	glucose catabolic process	4	0.037491393
GO:0009069	serine-family amino-acid metabolic process	4	0.037491393
GO:0019320	hexose catabolic process	4	0.037491393
GO:0006875	cellular-metal-ion homeostasis	3	0.037529777

**Table 2 jof-08-00111-t002:** KEGG enrichment of DEGs.

Pathway	Gene Number	*p*-Value	Pathway ID
2-Oxocarboxylic acid metabolism	14	0.001152635	ko01210
Lysine biosynthesis	6	0.001282578	ko00300
Valine, leucine, and isoleucine degradation	13	0.001659644	ko00280
Biosynthesis of antibiotics	57	0.001778891	ko01130
Biosynthesis of amino acids	30	0.006832061	ko01230
Nicotinate and nicotinamide metabolism	6	0.01082372	ko00760
Fructose and mannose metabolism	14	0.01136807	ko00051
Biosynthesis of secondary metabolites	69	0.01390241	ko01110
mRNA surveillance pathway	13	0.01922519	ko03015
Ribosome	21	0.02501491	ko03010
Metabolic pathways	155	0.02545925	ko01100
Mismatch repair	7	0.02694715	ko03430
Phenylalanine, tyrosine, and tryptophan biosynthesis	8	0.02985515	ko00400
Tyrosine metabolism	13	0.03127059	ko00350
Cysteine and methionine metabolism	13	0.03629479	ko00270
Valine, leucine, and isoleucine biosynthesis	6	0.03977772	ko00290

## Data Availability

Not applicable.

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
