# Peer review of "A Combined Analysis of Transcriptome and Proteome Reveals the Inhibitory Mechanism of a Novel Oligosaccharide Ester against Penicillium italicum"

_jof, 2022, doi:10.3390/jof8020111_

Round 1

Reviewer 1 Report

Review of “A combined analysis of transcriptome and proteome reveals the inhibitory mechanism of a novel oligosaccharide ester against Penicillium italicum

Basic reporting

This article agrees with Journal of Fungi scope.

The citrus spoilage by blue mold represents an important economic and health problem which is controlled my synthetic fungicides. Seeking natural bioactive compounds able to fight against mycotoxigenic molds is priority. In this work, the authors delve in understanding of Penicillium italicum transcriptomic and proteomic pathways triggered after it exposition to MTE-1, for which I consider the objective of this research to be widely justified and necessary.

The research has been conducted rigorously and methods are synthetically and detailed described or referenced.

Introduction is compact and aligned with the substance of work. Background and cited literature are sufficient and appropriate to frame the research.

The structure of the article is well organized according to the journal standard sections.

The manuscript is self-contained, and results are coherent with initial hypothesis. The structure guide properly to conclusions. Given the abundance of results and the complexity of interpreting them independently, a discussion as integrative and clarifying as the one that appears in this article is appreciated.

General comments for the author

Below are some specific suggestions to improve manuscript:

  1. Instead of introduction is sufficient, it would be improved including some information about how the oligosaccharide MTE-1 does its effect in other fruits and in which ones in particular.
  2. Is not clear for the reviewer if the oligosaccharide extracted from Pezicula neosporulosa has already described antifungal capacity.
  3. In in vivo treatment, MTE-1 is injected directly into the wound. In plate treatments, the effect seems to be due to direct oligosaccharide-mold contact. The concentrations in both cases are 0.4 g/l in high volumes in the case of the wound and, furthermore, as indicated, dose-dependent. How could it be applied as an effective and economically profitable post-harvest treatment? How?
  4. Results of Fig. 1 suggests that MTE-1 inhibits blue mold spore germination? Mold growth? Both of them? Reviewer wonder which is the perception of authors.
  5. Resolution in figures 1, 2 and 3 must be improved. Is quite difficult to appreciate mold infection in figure 3. Also in figure 2, the PDA plates are blurred.
  6. In figures 4c and 4d, cellular organelle should be pointed out.
  7. Line 252: lacks italic typography in “in vitro” and “in vivo”

Author Response

Basic reporting

This article agrees with Journal of Fungi scope.

The citrus spoilage by blue mold represents an important economic and health problem which is controlled my synthetic fungicides. Seeking natural bioactive compounds able to fight against mycotoxigenic molds is priority. In this work, the authors delve in understanding of Penicillium italicum transcriptomic and proteomic pathways triggered after it exposition to MTE-1, for which I consider the objective of this research to be widely justified and necessary.

The research has been conducted rigorously and methods are synthetically and detailed described or referenced.

Introduction is compact and aligned with the substance of work. Background and cited literature are sufficient and appropriate to frame the research.

The structure of the article is well organized according to the journal standard sections.

The manuscript is self-contained, and results are coherent with initial hypothesis. The structure guide properly to conclusions. Given the abundance of results and the complexity of interpreting them independently, a discussion as integrative and clarifying as the one that appears in this article is appreciated.

Response: We thank the reviewer so much for the positive comment.

 General comments for the author

Below are some specific suggestions to improve manuscript:

  1. Instead of introduction is sufficient, it would be improved including some information about how the oligosaccharide MTE-1 does its effect in other fruits and in which ones in particular.

Responses: We thank the reviewer for the valuable suggestion. MTE-1 is a novel compound. Currently, there are no reports on the effect of MTE-1 on postharvest disease of fruits. As suggested, we have made corresponding revision in the Introduction section in the revised manuscript.

  1. Is not clear for the reviewer if the oligosaccharide extracted from Pezicula neosporulosa has already described antifungal capacity.

Responses: Thanks for your careful review. Till now, MTE-1 has been only described with antifungal potential as we mentioned in reference (our patent) [11]. There are no reports on the effect of MTE-1 on postharvest disease of fruits. We have made corresponding revision in Introduction section in the revised manuscript.

  1. In in vivo treatment, MTE-1 is injected directly into the wound. In plate treatments, the effect seems to be due to direct oligosaccharide-mold contact. The concentrations in both cases are 0.4 g/l in high volumes in the case of the wound and, furthermore, as indicated, dose-dependent. How could it be applied as an effective and economically profitable post-harvest treatment? How?

Responses: In plate treatment, the inhibitory effect is related to both the concentration and the volume of the MTE-1. We revised the Figure 1A. MTE-1 was expressed as mass in Figure 1A, i.e. the filter paper discs (4 mm) contained 0, 6, 12 and 18 mg MTE-1. Figure 1B shows the inhibitory effect when 12 mg MTE-1 was used.

In practice, citrus fruit can be treated by dipping in 0.4 g/l MTE-1. We have evaluated the effect of MTE-1 on natural decay development in intact fruit by dipping fruit in 0.4 g/l MTE-1 and found that application of MTE-1 effectively reduced the decay development of harvested mandarins in semi-commercial experiments. Therefore MET-1 can be applied as an effective and economically profitable post-harvest treatment.

  1. Results of Fig. 1 suggests that MTE-1 inhibits blue mold spore germination? Mold growth? Both of them? Reviewer wonder which is the perception of authors.

Responses: Figure 1 shows that MTE-1 inhibits mold growth of Penicillium italicum

  1. Resolution in figures 1, 2 and 3 must be improved. Is quite difficult to appreciate mold infection in figure 3. Also in figure 2, the PDA plates are blurred.

Responses: Thank the reviewer for the suggestion. As suggested, we have made corresponding revision in the revised manuscript. ..

  1. In figures 4c and 4d, cellular organelle should be pointed out.

Responses: Thanks. We have made corresponding revisions in the revised manuscript. as suggested.

  1. Line 252: lacks italic typography in “in vitro” and “in vivo”

Responses: Thanks. We have made corrections in the revised manuscript.

Reviewer 2 Report

Dear Authors

This is an interesting study. The content of the paper could be published in the Journal of Fungi. But, the manuscript requires some minor reviews. In the manuscript, I pointed out some corrections. 

Author Response

This is an interesting study. The content of the paper could be published in the Journal of Fungi. But, the manuscript requires some minor reviews. In the manuscript, I pointed out some corrections.

Response: We thank the reviewer so much for the positive comment. As suggested, we have corrected the minor errors, including spelling, grammar. In addition, in Figure 2, the lowest concentration is 0.5 mg/L with the inhibition rate of 49.3%, not zero.